# Structural and Functional Characterization of the Holliday Junction Resolvase RuvC from *Deinococcus radiodurans*

**DOI:** 10.3390/microorganisms10061160

**Published:** 2022-06-06

**Authors:** Chen Qin, Wanchun Han, Ying Xu, Ye Zhao, Hong Xu, Bing Tian, Liangyan Wang, Yuejin Hua

**Affiliations:** MOE Key Laboratory of Biosystems Homeostasis and Protection, Institute of Biophysics, College of Life Sciences, Zhejiang University, Hangzhou 310058, China; 21916006@zju.edu.cn (C.Q.); hs19880208@163.com (W.H.); 11616018@zju.edu.cn (Y.X.); yezhao@zju.edu.cn (Y.Z.); xuhong1685@163.com (H.X.); tiangbing@zju.edu.cn (B.T.)

**Keywords:** Holliday junction, RuvC, *Deinococcus radiodurans*, Mn^2+^

## Abstract

Holliday junctions (HJs) are four-way DNA structures, which are an important intermediate in the process of homologous recombination. In most bacteria, HJs are cleaved by specific nucleases called RuvC resolvases at the end of homologous recombination. *Deinococcus radiodurans* is an extraordinary radiation-resistant bacterium and is known as an ideal model organism for elucidating DNA repair processes. Here, we described the biochemical properties and the crystal structure of RuvC from *D. radiodurans* (*Dr*RuvC). *Dr*RuvC exhibited an RNase H fold that belonged to the retroviral integrase family. Among many DNA substrates, *Dr*RuvC specifically bound to HJ DNA and cleaved it. In particular, Mn^2+^ was the preferred bivalent metal co-factor for HJ cleavage, whereas high concentrations of Mg^2+^ inhibited the binding of *Dr*RuvC to HJ. In addition, *Dr*RuvC was crystallized and the crystals diffracted to 1.6 Å. The crystal structure of *Dr*RuvC revealed essential amino acid sites for cleavage and binding activities, indicating that *Dr*RuvC was a typical resolvase with a characteristic choice for metal co-factor.

## 1. Introduction

Homologous recombination (HR) enables the cell to access and copy intact DNA sequence information in trans, which is an essential process in life [1]. In somatic cells, HR plays a key role in conserving genetic information by facilitating DNA repair. In higher organisms, HR is also involved in the meiosis that generates genetic diversity by reshuffling genes [2]. At the end of HR, after homologous pairing and strand exchange, two DNA double strands will form a four-way structure of DNA intermediates, called Holliday junctions (HJs), which must be removed by specific nucleases called resolvases at the end of the recombination process [3]. Resolvases from different biological kingdoms of life have significant diversity and belong to different classes of nucleases [4,5,6,7]. In eukaryotes, HJs are removed primarily by “dissolution”, a pathway involving the combined activities of a DNA helicase and a type IA topoisomerase, which catalyze branch migration and decatenation of the double HJ into non-crossover products [8,9]. In most bacterial cells, HJs are mainly processed by the RuvC resolvasome. The dimeric endonuclease RuvC symmetrically introduces two nicks at the junction, resulting in two separate recombinant DNA duplexes that can be directly repaired by DNA ligases [10,11,12]. In addition to RuvC, functional RuvA and RuvB are also needed for efficient Holliday junction resolution [13,14,15]. These proteins were initially identified by mutations that cause genetic defects in ultraviolet-induced DNA damage repair [16,17]. It is generally believed that these three proteins form a complex molecular machine called RuvABC lysosome, which coordinates the two main events of late recombination: (1) Two homologous duplex arms pass through the RuvA octamer to exchange their pairing partners [18]. (2) The RuvB complex functions as a pump to relocate the junction point to any cleavable sequences by pulling DNA duplex arms [11,12]. (3) RuvC catalyzes the decomposition of HJs via a pair of symmetrical incisions across the junction point [19].

*Deinococcus radiodurans* belongs to the *Deinococcus*-*Thermus* phylum and is highly resistant to various extreme environments and agents, including desiccation, ionizing radiation (IR), ultraviolet (UV) radiation and oxidative stress, thanks to its strong ability to repair DNA damage [20,21]. *Deinococcus* seldom invoke translesion synthesis and non-homologous end joining, but rather adopt homologous recombination to guarantee the fidelity of DNA repair [22]. Therefore, the efficient DNA damage repair ability of *D. radiodurans* may benefit from its powerful HR repair system. Despite the importance of RuvC in DNA recombination and repair across bacteria, there is essentially no biochemical and crystal study reported for RuvC from extremophiles. In this present study, we presented biochemical and structural analyses of RuvC from *D. radiodurans* (*Dr*RuvC) and demonstrated that Mn^2+^ rather than Mg^2+^ played key roles in the process of DNA binding and cleavage activities. This study will provide useful information for elucidating the efficiency of HR repair in *D. radiodurans*.

## 2. Materials and Methods

### 2.1. Oligonucleotides

All oligonucleotides and single nucleotide maker were purchased from Sangon (Shanghai, China). For DNA imaging, one strand of the double-helical DNAs was fluorescently labeled with 6-carboxyfluorescein (FAM). The sequences are listed in the Appendix A [23]. DNA annealing was carried out by mixing an equal molar amount of strands with complementary sequences in the annealing buffer (20 mM Tris-HCl pH 8.0, 50 mM NaCl), wherein the equal molar amount of strands with complementary sequences was mixed for annealing [24]. The mixture is heated at 98 °C for 5 min and then decreases by one degree per minute to room temperature.

### 2.2. Protein Expression and Purification

The gene encoding *Dr*RuvC was amplified by polymerase chain reaction and cloned into the expression vector *pET28a* between NdeI and BamHI sites. The encoded protein carries a 6 × His-tag sequence at the N-terminal. The constructed recombinant plasmid was transformed into BL21 (DE3) competent cells and cultured on LB plate containing 40 mg/L kanamycin. All the expressed strains were grown in LB medium containing 40 mg/L kanamycin at 37 °C to an optical density at 600 nm (OD_600_) of 0.6–0.8. Protein expression was induced at 30 °C for 5 h by adding 0.2 mM isopropyl β-D-1-thiogalactopyranoside (IPTG) [25].

After harvesting, cells were re-suspended in buffer A (20 mM Tris-HCl (pH 7.5), 500 mM NaCl, 10% (*v*/*v*) glycerol, and 1 mM β-mercaptoethanol), disrupted by sonication, and insoluble material was removed by centrifugation at 15,000 rpm for 35 min. All subsequent steps were performed at 4 °C. Furthermore, the supernatants were successively loaded onto nickel, desalting, ion exchange (Heparin, GE Healthcare, Pittsburgh, PA, USA), and size exclusion (Superdex™ 75/Superdex™ 200, GE Healthcare, Pittsburgh, PA, USA) columns using AKTA pure 25 (GE Healthcare, Pittsburgh, PA, USA) [26]. The columns and buffers used for each protein purification process were detailed in Appendix A. Finally, fractions were concentrated, aliquoted and stored at −80 °C in 50% glycerol. The purified protein was verified by SDS-PAGE (Figure 1A). The *Dr*RuvC mutants were expressed and purified similarly.

### 2.3. DNA Binding Assay

First, 50 nM of 5′-FAM labeled substrates were mixed with various amounts of *Dr*RuvC proteins to obtain a final volume of 20 μL, and the final solution conditions were 50 mM Tris-HCl, pH 7.0, 100 mM NaCl, 1 mM EDTA, and 1% glycerol. Binding reactions were incubated on ice for 30 min, then samples were electrophoresed through a 5% native polyacrylamide gel for 30 min at 200 V on ice. Gels were scanned by Typhoon FLA 9500 apparatus (GE Healthcare) [27].

### 2.4. DNA Cleavage Assay

For *Dr*RuvC enzyme nuclease activity assay, 100 nM *Dr*RuvC protein were mixed with 50 nM HJs in a 10 μL reaction mixture containing 50 mM Tris (pH 7.0), 100 mM NaCl, 1 mM MnCl2 and 0.1 mg/mL bovine serum albumin (BSA) at 37 °C for 5 min [27]. The reactions were terminated by the stopped solution containing protease K, SDS and EDTA. For divalent metal ion preferences of the nuclease activity assay, *Dr*RuvC and substrates were incubated with various concentration of MgCl_2_, MnCl_2_, CaCl_2_ or ZnCl_2_. The reaction mixture was separated on 5–10% polyacrylamide native gels, imaged and analyzed with Typhoon FLA 9500 apparatus (GE Healthcare). To ensure reproducibility, all assays were repeated at least three times independently.

### 2.5. Crystallization, Data Collection and Structure Determination

Purified *Dr*RuvC protein containing N-terminal 6 × His-Tag was concentrated to 3 mg/mL, and crystals were grown by the sitting-drop vapor diffusion method, using Index™-HR-144 Scoring Sheet (Hampton Research, Aliso Viejo, CA, USA) for initial screening at 293 K. *Dr*RuvC crystals were optimized and grown in the reservoir solution containing 0.2 M ammonium sulfate, 0.1 M BIS-TRIS pH 6.5, 25% *w*/*v* Polyethylene glycol 3350. Cryocooling was done by soaking the crystals in the reservoir solution supplemented with 20% (*v*/*v*) glycerol as cryoprotectant and fast-frozen in liquid nitrogen. Diffraction intensities were recorded on beam-line BL17U at the Shanghai Synchrotron Radiation Facility (Shanghai, China) and were integrated and scaled using the XDS suite [28,29]. The structure was determined by molecular replacement using a published *Ec*RuvC structure (PDB ID: 1HJR) as the search model. Structures were refined using PHENIX [30,31] and interspersed with manual model building using COOT [32]. Data collection and refinement statistics are summarized in Table 1. The coordinates and structure factors have been deposited to Protein Data Bank with accession codes 7XHJ. All structural figures in this study were generated with the PyMOL program. All figures were generated using the program PyMOL.

### 2.6. Circular Dichroism Spectroscopy

Circular dichroism spectra were obtained using a JASCO J-1500 spectrometer, equipped with a N_2_ purge and a Peltier system (PTC-4235) to control the temperature. The spectra were recorded between 200 nm and 260 nm. Measurements were carried out at 100 nm min^−1^ scan speed with a response time of 1 s and bandwidth of 1 nm. All the samples were measured in a precision cell made of Quartz Suprasil (Hellma) with a path-length of 1 mm. The intensities of the CD spectra were normalized as follows:(1)$$\theta =\frac{\theta_{\mathrm{measured}}\cdot M_{enzyme}}{10\cdot C_{\mathrm{RuvC}}\cdot l}$$
where *θ*_measured_ is the measured spectrum in mdeg, *l* is cuvette path-length, C_R__uvC_ is the concentration of *Dr*RuvC in the measured sample and Menzyme is the molecular weight of *Dr*RuvC.

## 3. Results

### 3.1. DrRuvC Is a Typical Holliday Junction Resolvase

Bioinformatics analysis showed that DR_0440 from *D. radiodurans* is a RuvC homolog (we designate it *Dr*RuvC) with a longer C-terminal tail (residues 163–179) compared with some homologous proteins (Appendix A). Both the full-length and the C-terminal tail truncated mutant *Dr*RuvC were overexpressed in the *E. coli* cells and purified. The full-length *Dr*RuvC protein was eluted as homodimer on the Superdex 75 size exclusion column (Figure 1A).

Previous studies have shown that Holliday junction resolvase recognizes substrates in a structure-specific manner. Sequence-dependent HJ resolution has been reported for RuvC (5′-A/TTT↓G/C-3′), Cce1 (5′-ACT↓A-3′), Ydc2 (5′-C/TT↓−3′), and MOC1 (5′-C↓C-3′) except the homologous proteins in viruses [23,33,34,35]. The DNA binding activities of *Dr*RuvC to the synthetic DNA substrates with various structures including duplex, splayed duplex, bulge, flap, Y-junction, HJ-0X and HJ-12X (Appendix A) were measured. It was demonstrated that HJs are the preferred substrate for *Dr*RuvC binding, and possesses the highest binding affinity, forming a stable DNA-protein complex band (Figure 1B). No stable shifted-bands were observed with the increase of *Dr*RuvC concentration in the presence of other DNA structures (Figure 1C). Consistently, *Dr*RuvC specifically cleaved the HJs, but not other DNA substrates (Figure 1D). To verify the sequence-specific cleavage ability of *Dr*RuvC for two kinds of Holliday junctions with different core sequences, HJ-0X and HJ-12X, were tested to verify the sequence specificity of *Dr*RuvC. HJ-12X has 12 bp homologous regions so that it can be migrated within a certain range, whereas HJ-0X is completely fixed and cannot be migrated (Appendix A). Native PAGE analysis showed that, *Dr*RuvC can effectively cleaved HJ-12X, but has no activity for HJ-0X (Figure 1D), which is similar to other RuvC homologues. However, *Dr*RuvC showed no significant difference in the binding of the two kinds of substrates (Figure 1B), suggesting that the homologous core can only affect the digestion ability of *Dr*RuvC, but not its binding ability. These data show that *Dr*RuvC functions as a typical HJ-DNA resolvase with strict specificity for the cleavage of branched DNA forms.

### 3.2. Mn^2+^ Is Essential for the Resolvase Activity of DrRuvC

Resolvase usually prefers Mg^2+^ as a co-factor for catalytic reactions, however, Mn^2+^ can act as a substitute [19,36,37,38]. To study the metal ion dependence of *Dr*RuvC, we used an HJ-12X substrate with a 12 bp homology region to react under different metal ion conditions. To create a contrast, 10 mM EDTA was included in the reaction buffer to sequester divalent metal ions. Interestingly, Mn^2+^ rather than Mg^2+^ is the only catalytic co-factor of *Dr*RuvC resolvase. *Dr*RuvC cleaved HJ-12X efficiently in the presence of Mn^2+^. However, no activity was detected in the presence of Ca^2+^, Zn^2+^, or Mg^2+^ (Figure 2A). Concentration-dependent experiments showed that no activity was detected with Mg^2+^ even up to 50 mM (Figure 2B). On the contrary, the DNA substrate was completely cleaved with Mn^2+^ at a concentration of 1 mM. We also found that 50 mM Mn^2+^ did not display any inhibitory effect on the activity of *Dr*RuvC (Appendix A), although studies have shown that high concentrations (>50 mM) of metal ions can inhibit the resolvase activity of other RuvC [10,37]. Since Mn^2+^ can enhance the digestion ability of *Dr*RuvC, we tried to perform HJ-0X cleavage assay in the presenc of Mn^2+^. Unfortunately, even a high concentration of Mn^2+^ cannot catalyze *Dr*RuvC to cleavage HJ-0X (Appendix A), suggesting that the homologous core is strictly needed in the enzyme digestion reaction.

### 3.3. High Mg^2+^ Concentration Inhibits the Binding Activity of DrRuvC

It is well-known that divalent metal ions are essential for the activity of resolvase, but not for DNA binding. We thus performed band-shift DNA binding analysis under different divalent metal ion conditions. To prevent DNA cleavage, we used HJ-0X without a homologous region to react. Similarly, we included 10 mM EDTA in the binding reaction buffer as a control. Native PAGE analysis showed that stable complexes were formed between *Dr*RuvC and HJ either in the absence of metal ions or the presence of Mn^2+^, Ca^2+^, and Zn^2+^. However, no shifted DNA appeared in the presence of Mg^2+^ (Figure 3A).

We further set a concentration gradient to detect the effect of Mg^2+^ and Mn^2+^ on DNA binding. The results showed that when the concentration of Mg^2+^ was higher than 1.5 mM, the shifted DNA band disappeared completely. However, all HJs were combined with *Dr*RuvC to form stable complexes with an increase in Mn^2+^ (Figure 3B). We also prepared a protein–DNA complex by adding different amounts of protein and a fixed concentration of DNA. We found that the binding ability of *Dr*RuvC to HJs in the presence of Mn^2+^ is much higher than that in the absence of metal ions (Figure 3C). Our results show that Mg^2+^ inhibits the binding of *Dr*DuvC to HJs, whereas Mn^2+^ facilitates the formation of the complex. This helps to explain the inhibitory effect of Mg^2+^ on the activity of *Dr*RuvC resolvase.

### 3.4. Crystal Structure of DrRuvC Reveals Its Essential Amino Acid Sites

The structure of *Ec*RuvC (PDB ID: 1HJR) was used as a search model, and the crystal structure of *Dr*RuvC was determined at 1.6 Å using the molecular replacement method. It contains two monomers in an asymmetric unit, and each monomer comprises 1–154 residues with an elongated overall shape. For each protomer, the protein folds of *Dr*RuvC consist of three antiparallel (β1β2β3) and two parallel strands (β4β5), sandwiched between five helices (α1–5), which is characteristic of enzymes from the retrovirus integrase superfamily (Figure 4A). Structural superimposition of *Dr*RuvC with *Ec*RuvC and *Tt*RuvC showed similar protomers (with an RMSD value of 1.656 Å) and dimeric architectures (Figure 4B), as expected from the high amino acid sequence identity between the two proteins. The structural deviations are mainly reflected in the secondary structural elements and the disordered region of the C-terminal.

The distribution of the electrostatic surface potential of *Dr*RuvC revealed that the catalytic center with a negatively charged groove is surrounded by a positively charged surface (Figure 4C), which corresponds to divalent metal ion and DNA binding sites. As expected for other known Holliday junction resolvases [4,18,36,39], the two catalytic centers in the *Dr*RuvC dimer are positioned on the same side composed of Asp7, Glu67, His139 and Asp142 (Figure 5A). The results of amino acids site mutation showed that the substitution of these amino acid residues can eliminate the activity of *Dr*RuvC completely (Figure 5B). Although these catalytic residues are conserved in most homologous proteins, the third metal-chelating residue of *Dr*DuvC is His143, corresponding to Asp138 of *Ec*RuvC (Figure 5C). Structurally, the pocket of the catalytic center is tighter in *Dr*RuvC than in *Tt*RuvC, which may be because Mn^2+^ is much smaller than Mg^2+^. It is worth noting that *Ec*RuvC does not choose Mg^2+^ as the only catalytic co-factor, although its third metal-chelating residue is His143.

Previous studies have revealed that the base-specific recognition loop (BR-loop, corresponding to E67-A75 of *Dr*RuvC) of MOC1 mediates its sequence-specific interaction with HJ [24]. As RuvC and MOC1 are evolutionarily conserved, we compared the BR-loop between RuvC proteins [40]. The results showed that the amino acid residues of *Dr*RuvC in the BR-loop were significantly different from that of other homologous proteins (Appendix A). Generally, aromatic residues such as tyrosine or phenylalanine are present within the loop. Following the aromatic residue (Tyr or Phe), there are charged (Asp, Lys, Arg) or polar (Asn, Gln, Ser) residues, which might mediate specific base recognition [24]. However, the BR-loop of *Dr*RuvC does not contain any amino acids with aromatic bases. In particular, the residue F69 in *Ec*RuvC is replaced by I70, which can intercalate into the bases one nucleotide before the cleavage site, forming stacking interactions with the flanking nucleotide bases [39,41,42]. The substitution of amino acids at these key sites may result in a distinctive mechanism of HJ cleavage by *Dr*RuvC.

## 4. Discussion

HR plays critical roles in repairing DNA damage including double-strand breaks [43]. RuvC proteins have been implicated as a key that mediates the decomposition of HJs at the end of HR [44]. As an important model organism, *D. radiodurans* is of great value in the study of DNA repair [45]. In this study, besides confirming the structure specificity of *Dr*RuvC, we explored the unique metal ion dependence of *Dr*RuvC. To gain insights into the cleavage mechanisms of RuvC, we determined the crystal structures of *Dr*RuvC and performed crystallographic analyses.

A panel of different branched DNA substrates was used to analyze the DNA binding and cleavage activities of *Dr*RuvC. Our data revealed the enzyme had the highest binding affinity for HJ DNA, consistent with the result that HJs, but no other substrates, were cleaved, suggesting that the recognition of DNA branches by the formation of the DNA–protein complex is the basis for determining the catalytic specificity of DNA substrates. Sequence-dependent cleavage by RuvC resolvases is another important property that needed to be addressed. Bacteriophage enzymes can cleave a broad range of branched DNA structures, including the HJ, with little sequence-specificity [33,35,46]. In contrast, sequence-dependent HJ resolution was reported in various cellular resolvases [19]. Two kinds of HJ substrates were used in our experiment: (1) HJ-12X with a 12 bp homologous region core and (2) HJ-0X with a non-homologous region core. The results suggested that the homologous core of the HJ is the basis for determining cleavage activity, but it does not affect the binding ability. The specific preference in the sequences of cleavage sites remains to be accurately analyzed.

Our results seem to show that *Dr*RuvC has a strict requirement for manganese as its metal co-factor. At present, all known RuvC proteins require divalent cations as catalytic co-factors for cleavage activity, among which Mg^2+^ is the most suitable [36,37,39], and Mn^2+^ could substitute for Mg^2+^ as a catalytic co-factor with weaker catalytic ability. However, it seems that *Dr*RuvC prefers Mn^2+^ as the only catalytic co-factor, whereas high Mg^2+^ concentration inhibits the binding between *Dr*RuvC and HJ. In addition, Mn^2+^ facilitates the formation of the *Dr*RuvC-HJ complex, although it is generally believed that this topology-specific binding does not require divalent cations. In fact, the similar phenomenon regarding the selection of Mn^2+^ and Mg^2+^ is widespread in *D**. radiodurans*. Compared with *E. coli* homologues that mainly use Mg^2+^ as cofactors, metalloenzymes in *D. radiodurans* exhibited a strong preference for Mn^2+^ rather than Mg^2+^. For example, Mn^2+^ are found in the structures of *Dr*Xth [26], RecJ [47], RNase J [48], MazG [49], MntH [50], and SodA [51] in *D**. radiodurans*. The circular dichroism (CD) results showed that the overall configuration of *Dr*RuvC protein did not change significantly under different metal ionic conditions (Appendix A). In the presence of metal ions, the helicity of protein decreased, which may be beneficial to the catalytic reaction of the protein [52]. In particular, Mg^2+^ makes *Dr*RuvC more relaxed than Mn^2+^, while less amount of α-helix conformation may hinder the enzyme digestion. *D. radiodurans* is known to accumulate very high intracellular manganese and low iron level compared to radiosensitive bacteria [53,54,55]. The high manganese concentration was suggested to be essential for relieving oxidative stress and protecting proteins from damage caused by reactive oxygen species [55,56]. The abundance of Mn^2+^ in the *D. radiodurans* cells may be used to explain why *Dr*RuvC has adopted Mn^2+^ as its metal co-factor.

The crystal structure of *Dr*RuvC shows an overall protein fold similar to that of *Ec*RuvC, but *Dr*RuvC has a tighter catalytic pocket by replacing the Asp138 of the catalytic active site with His139. In the base-specific recognition loop, the aromatic amino acids of *Dr*RuvC are replaced by leucine and isoleucine, indicating that *Dr*RuvC may have a specific cleavage mechanism. To determine the complex structure, a series of HJ with varying arm lengths were used to co-crystallize with *Dr*RuvC, but without success. The detailed structural mechanisms of metal ion preference and HJ resolution require further investigation. Notably, it has been recently proposed that the *Tt*RuvC resolvase could catalyze DNA cleavage through a general mechanism that is shared with CRISPR-Cas9 [57]. The DDE motif in Cas9 matches the Mn^2+^-coordinating ligands in *Dr*RuvC, and the catalytic residue H139 is located at H983 in Cas9. Although the metal-dependent mechanisms of the two enzyme digestion systems are different, it is worthy to probe the histidine-activated mechanism of *Dr*RuvC in recombination and genome editing.

## Figures and Tables

**Figure 1 microorganisms-10-01160-f001:**
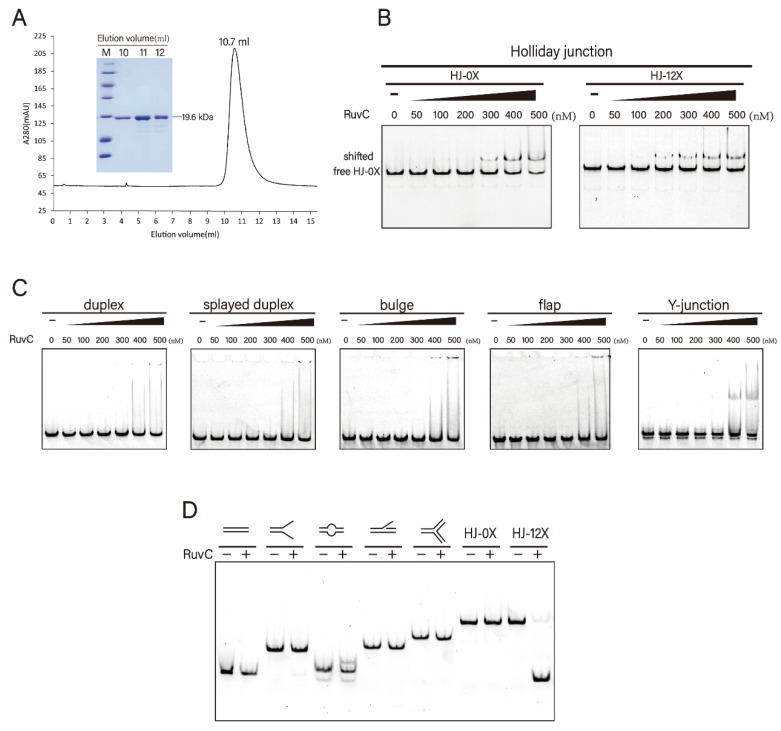
Protein purification and substrate specificities of *Dr*RuvC. (**A**) Size exclusion chromatogram (Superdex 75 10/300 GL column) of recombinant *Dr*RuvC protein. SDS-PAGE of representative eluted fractions is shown in inset. (**B**) Bandshift analysis of Holliday junctions (HJ-0X and HJ-12X). (**C**) Bandshift analysis of the indicated DNA structures. From left to right, the binding reactions contained 0, 50, 100, 200, 300, 400, 500 nM of *Dr*RuvC protein. (**D**) Native PAGE analysis of *Dr*RuvC cleavage of the indicated DNA structures.

**Figure 2 microorganisms-10-01160-f002:**
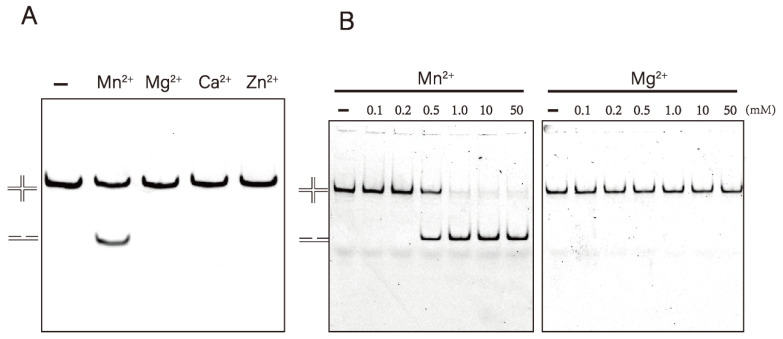
Effect of metal ions on the resolvase activity of DrRuvC. (**A**) The Holliday junction cleavage assays were performed in the absence or presence of indicated metal ions. Substrate uses HJ-12X with 12 bp homologous core. 500 μM of each metal ion was used. (**B**) Metal ion titration experiment of *Dr*RuvC cleavage activity.

**Figure 3 microorganisms-10-01160-f003:**
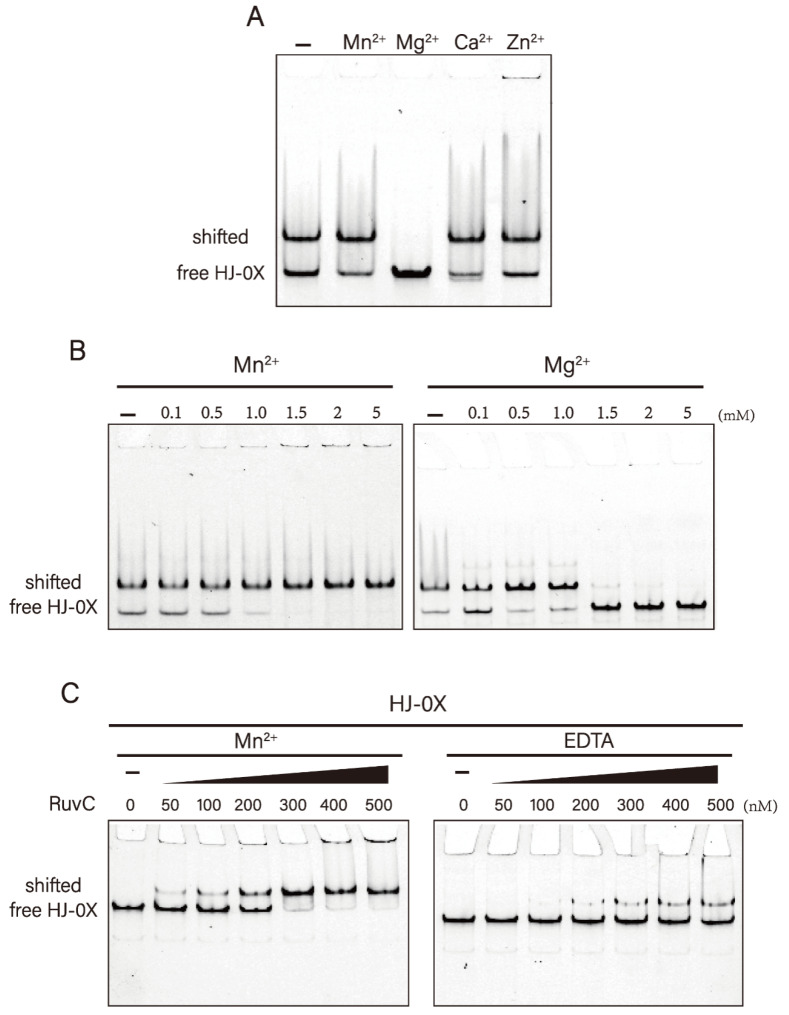
Effect of metal ions on the HJ binding of *Dr*RuvC. (**A**) The Holliday junction binding abilities of DrRuvC (250 nM) in the absence or presence of indicated metal ions (2 mM). The substrate HJ-0X with non-homologous core was used. (**B**) Mn^2+^ and Mg^2+^ titration experiments of HJ binding. (**C**) The binding abilities of DrRuvC to HJ-0X in the presence of 1 mM Mn^2+^ or in the absence of bivalent metal ions.

**Figure 4 microorganisms-10-01160-f004:**
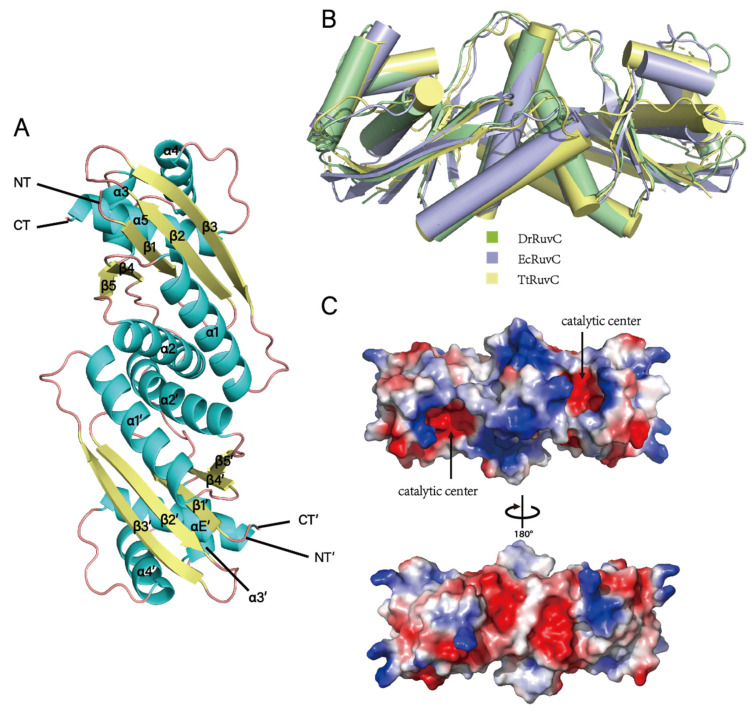
Crystal structure of *Dr*RuvC. (**A**) Cartoon diagram of the overall structure of *Dr*RuvC. The secondary structural elements are labeled. The N- and C-termini are indicated. (**B**) Structural superimposition of *Dr*RuvC with *Ec*RuvC and *Tt*RuvC. (**C**) Electrostatic surface of the *Dr*RuvC dimer colored according to the electrostatic surface potential (red: −1 kT/e to blue: +1 kT/e), shown in two orientations rotated by 180°. The location of the active sites is indicated by arrows.

**Figure 5 microorganisms-10-01160-f005:**
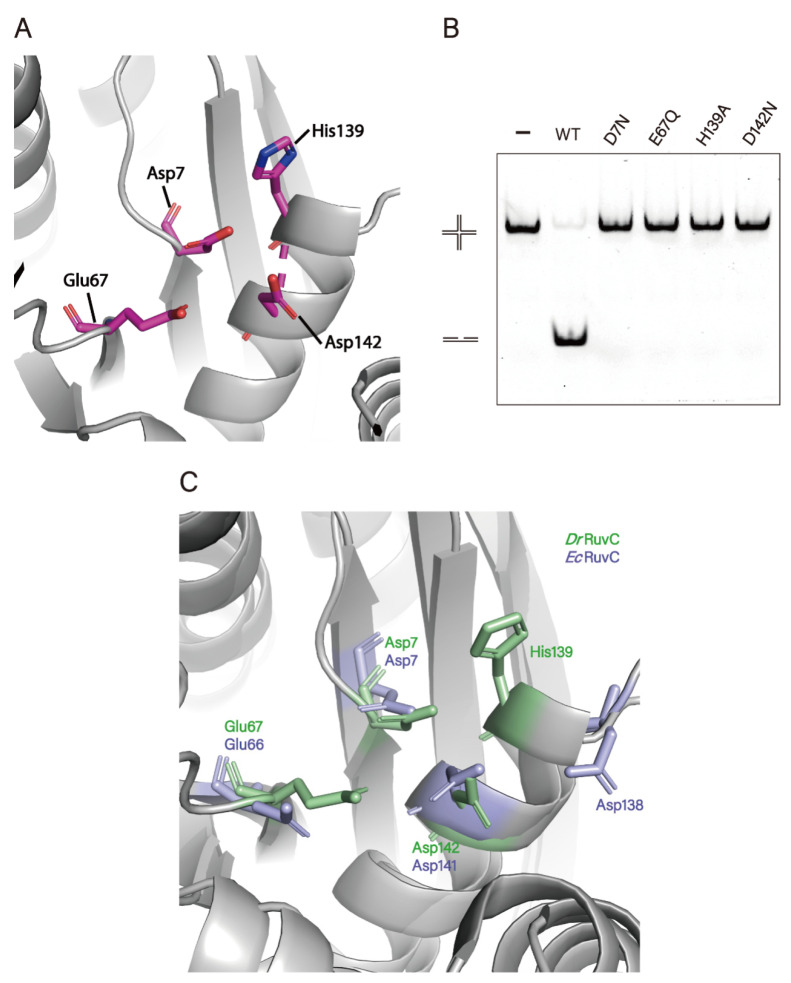
The active sites of *Dr*RuvC. (**A**) The active site of *Dr*RuvC. *Dr*RuvC is shown in grey and the catalytic residues are highlighted in magenta sticks. (**B**) Holliday junction cleavage assay of *Dr*RuvC wild type (WT) or various mutants. (**C**) Superposition of the catalytic residues from *Dr*RuvC (green) and *Ec*RuvC (blue).

**Table 1 microorganisms-10-01160-t001:** Data collection, phasing and refinement statistics.

	*Dr*RuvC
**Data collection**	
Space group	*P*2_1_2_1_2_1_
Cell dimensions	
*a*, *b*, *c* (Å) α, β, ɣ (°)	40.02, 72.60, 113.90 90, 90, 90
Wavelength (Å)	0.9793
Resolution (Å)	30.0–1.60
*R*_sym_ (%)	5.7 (41.1)
*I*/σ*I*	16.5 (4.3)
Completeness (%)	94.8(98.6)
Redundancy	3.7 (3.8)
**Refinement**	
Resolution (Å)	30.0–1.60 (1.64–1.60)
No. reflections	42,358
*R*_work_/*R*_free_	20.5/22.4
No. atoms	
Protein	2270
Water	151
B-factors	
Protein	30.7
Water	42.5
R.m.s deviations	
Bond lengths (Å)	0.005
Bond angles (°)	0.770
Ramachandran statistics	
Favored (%)	98.7
Allowed (%)	1.3
Ourlier (%)	0

## Data Availability

The coordinates and structure factors have been deposited to Protein Data Bank with accession codes 7XHJ.

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
