# Peer review of "Structural and Functional Characterization of the Holliday Junction Resolvase RuvC from Deinococcus radiodurans"

_microorganisms, 2022, doi:10.3390/microorganisms10061160_

Round 1

Reviewer 1 Report

The authors propose the X-ray structure of a RuvC resolvase enzyme. The paper is well written and the data are consistent, in line with previous findings. The paper requires moderate revisions in light of what is known on RuvC enzymes. It has been recently proposed that the RuvC resolvase could catalyze DNA cleavage through a general mechanism that is shared with CRISPR-Cas9. Based on the catalytic mechanism of RuvC in CRISPR-Cas9, it has been proposed that the mechanism is activated by a catalytic histidine and is shared by enzymes performing genome editing and recombination (Casalino et al. ACS Catalysis, 2020, 22, 13596-13605). The authors should discuss the catalytic activation of RuvC in light of the findings on CRISPR-Cas9.  

Reviewer 2 Report

Review comments:

In this manuscript, Qin et.al made a functional characterization of the holliday junction resolvase Ruvc from Deinococcus radiodurans bacteria which possess strong DNA damage repair ability.  The author reports the substrate selection and ion co-factor influence of the DrRuvC enzymatic activity. The author also solved the apo form structure of DrRuvC and try to analysis DrRuvC from structural perspective.  The data is not that sufficient.  I would endorse this paper for publication after following points to be addressed:

The key novelty of this paper is they indicate Mn2+ is the active ion in DrRuvC and Mg2+ inhibit the substrate binding at high concentrate. But the author didn’t make a reasonable explanation from both biochemistry and structural perspectives.  The author may claim the ion on the protein binding and activity in different points: 1) when secondary structure changed in high Mg2+, for example CD results of the protein in different ion-containing buffer; 2) an ITC or other binding verification experiment to check the binding ability of the Mg2+, Mn2+ with protein; 3) If all above show no big difference, then it might because of Mg2+ can occupy the active site as Mn2+ but it cannot transfer electron when it function enzymatic activity in the reaction. The surrounding residues responsible for ion binding and electron transfer is different in the structure.  The author may not need do all the experiment above. But they should try to claim the biochemistry or molecular mechanism of this novelty point of this paper.

Major comments:

  1. The table 1 is not sufficient structure determination using X-ray crystallography method.

Please revise carefully about the listed points:

  • What’s the angle information in the cell dimensions?
  • What’s the outer shell information of the resolution? The resolution indicate here is 1.6 Å while it was noted 1.7 Å in the main text.
  • It seems the author was not made a sufficient refinement of this dataset. The Rwork/Rfree and the B-factor is relative higher at this resolution though further refinement won’t influence the overall structure of the molecular.
  • The information of Ramachandran plot is missing.
  1. In 3.1 first paragraph. The author claimed the special longer C-terminal in DrRuvC based on the sequence alignment. While the alignment is not sufficient, for example streptomyces griseus also contain longer C-terminal. Besides, what’s kind of stability if C-terminal removed. Please show details and results if you want to claim this point in the text.
  2. It’s very interesting that HJ-0X can be bound while cannot be cleaved by Dr It seems HJ-0X owns higher binding ability comparing with HJ-12X. As reported, RuvC is sequence dependent for its enzymatic activity. In the indicated figure S3, HJ-0X and HJ-12X , there is no (A/T)TT ↓(G/C)  cleavage sites.  Could you highlight the cleavage site in the HJ-DNA in the Fig.S3. Since the Mn2+ can enhance the digestion ability of DrRuvC. Would it be possible for the cleavage of HJ-0X at different Mn2+ containing buffer condition?
  3. Figure 2A, 500nM of each metal used? Is 500nM or 500uM. No activity at 0.1mM for Mn2+ while obvious activity in Figure 2A for Mn2+. Please Check the concentration of each digestion and binding experiment. eg. In the Figure 3C, the Mn2+ is 1nM? It’s confusing.
  4. The author undertook the resolvase activity and binding activity under various ion concentration. In result 3.2, the author claimed higher concentration didn’t inhibit the DrRuvC activity. The author should provide a time-resolved cleavage data if they try to claim this point.
  5. In result 3.4, what method or website used for structural superposition? The RMSD of the monomer superposition should be indicate at the same time. I prefer the author make a superposition of DrRuvC with TtDuvC (apo/DNA-bound) at the same time.
  6. In result 3.4, the author tried to analysis the pocket of the catalytic center in DrRuvC comparing with TtRuvC. Please provide analysis images. “Mn2+” is much smaller than “Mg2+”. Is the correct? I prefer the author make a detail sequence and structural analysis of DrRuvC with currently well know EcRuvC and TtRuvC to try to explain the substrate binding and ion-selection mechanism. Besides, a recovery assay maybe be done to verify their analysis. For example, if they believe His143 is key point for Mn2+ selection comparing this in Asp138 in EcRuvC. They can mutate these residues in Asp to check the activity. 
  7. Have the authors made a soaking of Mn2+ in the crystal and solve the ion-binding structure? This may be not a hard experiment to try.
  8. The author should cite papers sufficiently. Please check in the whole manuscripts including the method parts.

Round 2

Reviewer 2 Report

Review comments:

Overall summary(2nd):

As I mentioned in the first revision, the novelty of this manuscript is limited. Thus, I hope the author can carefully revised and take sufficient experiment and analysis to support their identified special points as to DrRuvC. For example, the Mn2+ selectivity for enzymatic activity. The new revision made a great progress and make it much clearer, while there are still some confusing questions to be answered. I hope the author do not ignore them automatically.  I attached the first review opinions and answers at the same time for better explain:

In this manuscript, Qin et.al made a functional characterization of the holliday junction resolvase Ruvc from Deinococcus radiodurans bacteria which possess strong DNA damage repair ability.  The author reports the substrate selection and ion co-factor influence of the DrRuvC enzymatic activity. The author also solved the apo form structure of DrRuvC and try to analysis DrRuvC from structural perspective.  The data is not that sufficient.  I would endorse this paper for publication after following points to be addressed:

The key novelty of this paper is they indicate Mn2+ is the active ion in DrRuvC and Mg2+ inhibit the substrate binding at high concentrate. But the author didn’t make a reasonable explanation from both biochemistry and structural perspectives.  The author may claim the ion on the protein binding and activity in different points: 1) when secondary structure changed in high Mg2+, for example CD results of the protein in different ion-containing buffer; 2) an ITC or other binding verification experiment to check the binding ability of the Mg2+, Mn2+ with protein; 3) If all above show no big difference, then it might because of Mg2+ can occupy the active site as Mn2+ but it cannot transfer electron when it function enzymatic activity in the reaction. The surrounding residues responsible for ion binding and electron transfer is different in the structure.  The author may not need do all the experiment above. But they should try to claim the biochemistry or molecular mechanism of this novelty point of this paper.

Response 1: Thank you for the suggestion. We have carried out the CD assay and obtained the data of the DrRuvC protein in different ion-containing buffer. The result has been shown in Supplementary Fig. S6, while the related information has been incorporated into Discussion in the revised manuscript. “The circular dichroism(CD) results showed that the overall configuration of DrRuvC protein did not change significantly under different metal ionic conditions (Supplementary Fig. S6). In the presence of metal ions, the protein seems to transit from a relaxed state to a more compact state, which may be beneficial to the catalytic reaction of the protein. In particular, Mg2+ makes DrRuvC more compact than Mn2+, while excessive compacting may hinder the enzyme digestion.”

2nd round review: Please including the experiment details of CD experiment in the method part.

What I got from the curve is the with additional Mn2+ and Mg2+ added, the helicity of the protein decreased. Could the author including the reference paper for the conclusion “ In the presence of metal ions, the protein seems to transit from a relaxed state to a more compact state, which may be beneficial to the catalytic reaction of the protein” for the explanation of CD curve.

I should mention again: The key novelty of this paper is they indicate Mn2+ is the active ion in DrRuvC and Mg2+ inhibit the substrate binding at high concentrate. But the author didn’t make a reasonable explanation from both biochemistry and structural perspectives.  The author may claim the ion on the protein binding and activity in different points. CD maybe not a good method from result indicated. What I expect here is the author try to explain the Mn2+ is the active ion for this special DrRuvC from biochemistry or structure perspective. This is the novelty of this paper.

Major comments:

1.     The table 1 is not sufficient structure determination using X-ray crystallography method.

Please revise carefully about the listed points:

1)    What’s the angle information in the cell dimensions?

2)    What’s the outer shell information of the resolution? The resolution indicate here is 1.6 Å while it was noted 1.7 Å in the main text.

3)    It seems the author was not made a sufficient refinement of this dataset. The Rwork/Rfree and the B-factor is relative higher at this resolution though further refinement won’t influence the overall structure of the molecular.

4)    The information of Ramachandran plot is missing.

Response 2: Thank you for careful revision and valuable suggestion. We have supplemented the data in Table 1. The value of resolution is 1.6 Š,and we have corrected the mistakes in the maintext. Although the value of Rwork/Rfree is a little high, it is still within the normal range. For more specific information, please see the uploaded attachment “Full wwPDB X-ray Structure Validation Report”

2nd round review: in 1st review comment. What’s the outer shell information of the resolution? The author still didn’t show in the table.  Since the author show the information of Rsym, Redundancy et.al of the out-shell, the resolution should be indicated at the same time. The author response: “Although the value of Rwork/Rfree is a little high, it is still within the normal range.” Generally, for the resolution at 1.6, the Rwork is around 16. Besides, it also depends on how much percentage of reflections used to calculate Rfree.  

The reason I strongly suggest the author make a carefully refinement is because this is an enzyme with special ion binding site. Carefully refinement influence the information get from the active site. Please take this suggestion and take a further look at your structure.

2.     In 3.1 first paragraph. The author claimed the special longer C-terminal in DrRuvC based on the sequence alignment. While the alignment is not sufficient, for example streptomyces griseus also contain longer C-terminal. Besides, what’s kind of stability if C-terminal removed. Please show details and results if you want to claim this point in the text.

3.     It’s very interesting that HJ-0X can be bound while cannot be cleaved by DrRuvC. It seems HJ-0X owns higher binding ability comparing with HJ-12X. As reported, RuvC is sequence dependent for its enzymatic activity. In  the indicated figure S3, HJ-0X and HJ-12X , there is no (A/T)TT (G/C)  cleavage sites.  Could you highlight the cleavage site in the HJ-DNA in the Fig.S3. Since the Mn2+ can enhance the digestion ability of DrRuvC. Would it be possible for the cleavage of HJ-0X at different Mn2+ containing buffer condition?

Response 4: Thank you for the suggestion. We have tried to use the FAM-labled HJ to determine the exact cutting site of DrRuvC, but failed. Isotope labeling experiments are needed to clarify the cleavage sites. We have added this information in the maintext and Supplementary Fig. S4. “Unfortunately, Mn2+ can not catalyze DrRuvC to cleavage HJ-0X, suggesting that the homologous core is strictly needed in the enzyme digestion reaction. ”

2nd round review: So this is different from other general RuvC at the no (A/T)TT (G/C)   cleavage sites, am I right? If right, this is also a novelty of the paper and what the activity towards the  generally used (A/T)TT (G/C) HJ DNA. What’s the start consideration for you choosing HJ-12X, is there any other paper used or other reason?

4.      Figure 2A, 500nM of each metal used? Is 500nM or 500uM. No activity at 0.1mM for Mn2+ while obvious activity in Figure 2A for Mn2+. Please Check the concentration of each digestion and binding experiment. eg. In the Figure 3C, the Mn2+ is 1nM? It’s confusing.

5.     The author undertook the resolvase activity and binding activity under various ion concentration.  In result 3.2, the author claimed higher concentration didn’t inhibit the DrRuvC activity. The author should provide a time-resolved cleavage data if they try to claim this point.

Response 6: Thank you for the suggestion. We have designed and carried out experiments to verify the protein activity under the condition of high concentration of Mn2+, and the results have been shown in Supplementary Fig. S5.

2nd round review: a combine time curve based on the bands density of these results together may make it much clearer.

6.     In result 3.4, what method or website used for structural superposition? The RMSD of the monomer superposition should be indicate at the same time. I prefer the author make a superposition of DrRuvC with TtDuvC (apo/DNA-bound) at the same time.

Response 7: Thank you for the suggestion. PyMOL was used for structural superposition. The superposition of DrDuvC with EcDuvC and TtDuvC has been perfomed, while the RMSD of the monomer superposition has been indicated.

2nd round review: It’s Okay for using Pymol for structural superposition. DALI may be another better choice for superposition. The author can keep current result if there is no big difference.

7.     In result 3.4, the author tried to analysis the pocket of the catalytic center in DrRuvC comparing with TtRuvC. Please provide analysis images. “Mn2+” is much smaller than “Mg2+”. Is the correct?   I prefer the author make a detail sequence and structural analysis of DrRuvC with currently well know EcRuvC and TtRuvC to try to explain the substrate binding and ion-selection mechanism. Besides, a recovery assay maybe be done to verify their analysis. For example, if they believe His143 is key point for Mn2+ selection comparing this in Asp138 in EcRuvC. They can mutate these residues in Asp to check the activity. 

Response 8: Thank you for the comment. We compared the catalytic centers of DrRuvC and TtRuvC, and found that the difference is not obvious, suggesting that it may not be the main reason for ion preference, so we delete this inforamation in this paper. It’s a good idea to execute recovery assays. Unfortunately, we failed to obtain highly purified EcRuvC, therefore unable to conduct comparison assays.

2nd round review: “Unfortunately, we failed to obtain highly purified EcRuvC, therefore unable to conduct comparison assays.”  “For example, if they believe His143 is key point for Mn2+ selection comparing this in Asp138 in EcRuvC. They can mutate these residues in Asp to check the activity. ”

 My suggestion is how about  the result of His143 to Asp in DrRuvC, not mutantion from Asp to His in EcRuvC.   Still, the novelty of this paper is the mechanism of Mn2+ for the activation of this enzyme, any evidence will make this paper be better.

8.     Have the authors made a soaking of Mn2+ in the crystal and solve the ion-binding structure? This may be not a hard experiment to try.

Response 9: Thank you for the suggestion. We tried to add different metal ions to the crystallization buffer, but only the crystal of apo was obtained. Perhaps the crystallization conditions need to be further optimized.

2nd round review: My suggestion is soaking based on the ready crystal not co- crystallization. The author can still try and I hope there is good news.
